# Genomic Consideration in Chemotherapy-Induced Ovarian Damage and Fertility Preservation

**DOI:** 10.3390/genes12101525

**Published:** 2021-09-28

**Authors:** Seongmin Kim, Sanghoon Lee, Hyun-Tae Park, Jae-Yun Song, Tak Kim

**Affiliations:** 1Gynecologic Cancer Center, CHA Ilsan Medical Center, CHA University College of Medicine, 1205 Jungang-ro, Ilsandong-gu, Goyang-si 10414, Korea; naiad515@gmail.com; 2Department of Obstetrics and Gynecology, Korea University College of Medicine, 73 Inchon-ro, Seongbuk-gu, Seoul 02841, Korea; cyberpelvis@korea.ac.kr (H.-T.P.); yuni105@korea.ac.kr (J.-Y.S.); tkim@kumc.or.kr (T.K.)

**Keywords:** chemotherapy, gene, mutation, gonadotoxicity, fertility preservation, cryopreservation, oncofertility

## Abstract

Chemotherapy-induced ovarian damage and fertility preservation in young patients with cancer are emerging disciplines. The mechanism of treatment-related gonadal damage provides important information for targeting prevention methods. The genomic aspects of ovarian damage after chemotherapy are not fully understood. Several studies have demonstrated that gene alterations related to follicular apoptosis or accelerated follicle activation are related to ovarian insufficiency and susceptibility to ovarian damage following chemotherapy. This may accelerate follicular apoptosis and follicle reservoir utilization and damage the ovarian stroma via multiple molecular reactions after chemotherapy. This review highlights the importance of genomic considerations in chemotherapy-induced ovarian damage and multidisciplinary oncofertility strategies for providing high-quality care to young female cancer patients.

## 1. Introduction

It is estimated that 9.2 million women were newly diagnosed with malignancy worldwide in 2020 [1]. Among adolescents and young adults aged 15–39 years, 89,500 patients were newly diagnosed with cancer, and 9270 mortalities were reported in the United States [2]. In these patients, oncologic therapies can harm normal ovarian function and result in ovarian damage [3]. Fertility preservation is now an emerging discipline that plays a critical role in preventing infertility in the care of young cancer patients [4,5].

Chemotherapy could harm gonadal function in young cancer patients and cause loss of the ovarian reserve [6]. The molecular mechanism of chemotherapy-induced ovarian damage has been investigated to understand and prevent gonadotoxicity in cancer treatment [7]. However, the genetic aspects of chemotherapy-induced ovarian damage are still not fully understood. This article reviews the genetics of chemotherapy-induced ovarian dysfunction and explores the gene-targeted prevention of ovarian damage.

## 2. Genes Involved in the Regulation of Ovarian Follicular reserve

In females, number of primordial follicles (PFs) declines towards menopause because of their finite nature [8]. In every mammalian species, the ovarian reserve is formed early in life and then declines regularly throughout life [9]. It is created during ovarian histogenesis by follicular endowment [10]. Non-growing follicle endowment is composed of the formation, commitment, migration, and colonization of ovarian primordial germ cells (PGCs). The development of the bipotential gonad is followed by sex determination, definitive ovarian histogenesis, and follicular assembly [11,12]. Figure 1 shows the morphogenesis of follicles from the arrival of primordial germ cells to secondary follicles. The processes and related genes are listed in Table 1 [13,14,15].

### 2.1. Primordial Germ Cells Formation and Gonad Colonization

Several studies in mice have investigated important signals for primordial germ cell specification, migration, and proliferation. The deletion of *Bmp2*, *Bmp4*, *Bmp8a*, and downstream mediators such as *Smad* genes is related to the failure of migration or absence of PGCs [16,17,18,19,20]. *Oct4* expresses critical survival factors and forms pluripotent stem cells [21]. *Nanos* genes also have a specific role in the migration and proliferation of PGCs. *Nanos1* ablation was related to the failure of PGC migration, and *Nanos3* ablation caused PGC migration and proliferation defects [22,23]. *Kitl*, *Pin1*, and *Pog* are also known to play critical roles in the survival and establishment of PGCs [24,25,26].

### 2.2. Germ Cell Survival and DNA Damage Repair

Autophagy has a vital role in the regulation of follicle development. *Atg7* and *Becn1*, which are autophagic factors, are involved in follicular formation [27]. *Gja1* encodes connexin 43, which forms gap junctions between cells in the ovigerous cords, and plays a role in PGC development [28]. Genes such as *Spo11*, *M**sh4*, and *M**sh5*, which are involved in repairing DNA double-strand breaks (DSBs), may affect fertility and induce ovarian insufficiency [29,30,31]. The DNA strand-related gene *Dmc1* is responsible for maintaining follicular count [32]. Another DNA DSB-related gene is *Atm*, which is related to the loss of follicles and ovarian dysgenesis [33]. The role of *Brca1/2*, which is critical in homologous recombination to repair DSBs, in ovarian dysfunction is still under debate. Although several clinical studies have shown poor ovarian reserve in *Brca1/2* carriers, the exact role of *Brca1/2* in the ovarian reserve has not yet been determined [34,35].

Additionally, a loss of follicles is observed after *Rec8* ablation, which is a component of the cohesion complex [36]. Loss of *Cdk2*, which is involved in cell cycle progression, or *Cbep*, which regulates the synaptonemal complex, is related to germ cell loss [37,38]. *Fanc* family genes encode proteins that interact to mediate DNA damage repair [39]. Mutations in *Fanc* family genes may induce Fanconi anemia, thus also contribute to impairment of follicular development [40,41,42]. *Hsf1* induces the expression of heat shock proteins and initiates oocyte development. *Hsf1* is also responsible for the anti-oxidative stress in oocytes [43,44]. *Syce1* is essential for the formation of synaptonemal complexes [45]. Other meiotic gene mutations, including *Stag3*, *Pof1b*, *Pof2b*, and *Hfm1*, have also been associated with ovarian reserve impairment in humans [46,47,48].

### 2.3. Follicular Assembly and Turnover

Non-growing follicular assembly occurs with the degeneration of other oocytes surrounded by squamous pre-granulosa cells and the basement membrane. Mutations in several genes involved in this process could alter the ovarian reserve [15]. *Figla*, an oocyte-specific transcription factor, is necessary to form primordial follicles. Ablation of this gene results in failure of follicle formation [49]. When this gene is altered, ovarian insufficiency can occur [50]. Neurotrophins are a family of growth factors that regulate cell survival and follicular development. For example, *Ngf*, *Ntrk1*, and *Ntrk2* affect primordial follicle formation in mice [51]. Additionally, *Nt4* and *Bdnf* are related to follicular assembly and survival [52]. *N**t3* and *Ntrk3* also participate in the transition of follicles from the PF to the primary stage [53].

Apoptotic pathway-related genes are also related to follicle turnover. Alterations in *Casp2* and *Bcl2* have been shown to decrease the number of PFs [54,55]. *Ahr* and *Bax* play important roles in follicle maturation and PF endowment. *Bax* deletion is associated with the formation of a better ovarian reserve [56]. However, contradictory results have been reported in the literature [57]. Deletion of *Ahr,* whichs is a regulator of *Bax* expression, results in an increased PF count [58]. *Mcl1* is another gene involved in apoptosis. *Mcl1* deletion elevated superoxide levels and activated the autophagic pathway, reducing the ovarian reserve [59].

## 3. Mechanism of Chemotherapy-Induced Ovarian Damage

Chemotherapy-induced ovarian damage may be transient, and menstruation may recover after treatment completion. Oocytes and granulosa cells are vulnerable to chemotherapeutic agents. The possible gonadotoxic chemotherapeutic agents used are shown in Table 2 [7]. Each agent has a different mechanism of action on malignant cells, resulting in the cessation of the cell cycle. With conventional chemotherapy agents, ovarian insufficiency involves PF pool depletion by apoptosis or hyperactivation mechanisms, mediated by the ABL/TAp63 and PI3K/Akt/mTOR pathways [7].

### 3.1. Chemotherapy-Induced DNA DSBs

Chemotherapy can result in DSBs in DNA that can be repaired by the ataxia-telangiectasia mutated-mediated DNA damage repair pathway. However, failure of the repair pathway results in cellular apoptosis in growing follicles and proliferating granulosa cells [60]. P63 protein, a transcriptional factor implicated in cancer and development, is also involved in female reproduction [61]. TAp63, which is the N-terminal transactivation domain containing isoform of P63, is responsible for the protection of the female germ line during meiotic arrest [62]. The P63 protein activates BAX and BAK proteins, which can be transmitted by activating Tap73, a P53-upregulated modulator of apoptosis [63]. This damage has been reported to occur even with low-risk gonadotoxic agents [64].

### 3.2. Burnout Effect

The PI3K/Akt/mTOR pathway directly influences the oocytes and pre-granulosa cells of PFs and indirectly destroys large follicles, called the “burnout effect” [65]. This phenomenon impairs anti-Mullerian hormone (AMH) and reduces the suppression of the PF pool through destroying follicles, which is followed by the activation of PFs to compensate for the decrease in the number of growing follicles [66]. This effect triggers the growth of dormant follicles. It is affected by the upregulation of the PI3K/Akt/mTOR pathway and substantial follicular apoptosis, which reduces AMH secretion [67].

### 3.3. Stromal and Microvascular Damage

The ovarian stroma can be indirectly damaged by chemotherapeutic agents [8,68]. A previous study reported chemotherapy-induced ovarian stromal fibrosis and vascular damage [69]. Damage to blood vessels and focal fibrosis of the ovarian cortex could be another mechanism of chemotherapy-induced ovarian dysfunction [70]. In patients undergoing chemotherapy, the ovaries show thickening and hyalinization of the cortical vessels [71]. This is also supported by another study that showed an inverse correlation between ovarian vascular density and follicular apoptosis [72], thus suggesting an indirect mechanism by which chemotherapy-induced ovarian vascular injury reduces the number of PFs.

### 3.4. Genes Related to Chemotherapy-Induced Ovarian Damage

#### 3.4.1. DNA Damage Repair

Homologous or non-homologous DNA repair is involved in the recovery of chemotherapy-induced DNA damage in PFs. Consequently, mutations in genes that regulate these repair pathways could increase the susceptibility to ovarian toxicity due to chemotherapy.

*Brca1* and *Brca2* are critical in the repair of DNA DSBs. *Brca* mutation carriers have not only increased the risk of cancer but also fertility-related issues [73]. *Brca1* mutation carriers show lower AMH levels, but the results are contradictory between studies [74,75]. *Brca2* mutations are not associated with a low ovarian reserve in these studies. On the other hand, a retrospective study on the in vitro fertilization of *Brca* mutation carriers showed no significant differences in the procedure cycles or in the number of oocytes compared to non-carriers [76]. Additional research is warranted to define exact role of Brca mutation in fertility preservation in patients with related malignancy. In cancer patients with *Brca* mutations, poly (ADP-ribose) polymerase (PARP) inhibitors is widely used for the treatment of cancer [77]. The use of PARP inhibitors could negatively affect embryo development [78]. In another study, the gene expression of granulose cell markers was decreased in patients with PARP inhibitor use [79].

Alterations of other genes involved in DNA repair, *Mcm8* and *Mcm9*, can induce primary ovarian insufficiency [80]. *Stag3*, a meiosis-specific gene, is also important in DNA damage repair. A recent study demonstrated that variants of *Stag3* are associated with primary ovarian insufficiency [81]. Similarly, *Hfm1*, *Nup107*, and *Syce1* are associated with DNA repair and are implicated in ovarian insufficiency [45,82,83].

#### 3.4.2. Apoptosis

Dysregulation of apoptosis results in decreased ovarian reserve and an increased possibility of gonadal damage after chemotherapy. *Nanos3*, which expresses an RNA-binding protein that regulates apoptosis to maintain a proper PF pool, was related to ovarian insufficiency in a study of Chinese women with variant mutations [84]. In that study, the level of NANOS3 protein was correlated with the number of PGCs. Ablation of another important anti-apoptotic gene, *Bcl2*, is related to a decreased number of PGCs in mice [55]. *Pgrmc1*, which is another candidate gene, has a progesterone-dependent anti-apoptotic action, which is another candidate gene. Mutations in this gene were related to ovarian insufficiency in a previous study [85,86].

#### 3.4.3. Follicular Activation and Development

The possibility of ovarian damage after chemotherapy could also be increased because of genetic mutations involved in follicular activation and development. *Foxo3a* inhibits follicular activation in the ovary. Ablation of this gene in mice is related to early ovarian dysfunction [87]. In humans, *Foxo3a* and *Foxo1a* were identified in women with primary ovarian insufficiency in two studies [88,89]. Variants of another follicle developing gene, *Bmp15*, are associated with ovarian dysfunction, as identified in multiple studies [90,91,92].

## 4. Prevention Strategy for Ovarian Damage

Fertility preservation options can be personalized in terms of patient age, desire for conception, treatment regimen, and socioeconomic status [93]. Such options include hormonal medications for ovarian suppression, cryopreservation, in vitro oocyte maturation, artificial ovaries, and stem cell technologies. Additionally, the potential ovarian protective effects of several genetic variants could be considered. Several established options including embryo cryopreservation and oocyte cryopreservation are already in clinical use. However, there are also experimental options including ovarian tissue cryopreservation, oocyte in vitro maturation, artificial ovary, and stem cell technologies [93].

### 4.1. Consideration for Protective Genetic Variants for Chemotherapy-Induced Ovarian Damage

Several reports have been published regarding the protective effect of gene mutations associated with a better prognosis in terms of ovarian insufficiency. A protective effect of reduced allele frequency of the *Inha* gene promoter was observed in patients with premature ovarian insufficiency [90,91]. In a study involving ovarian insufficiency, increased expression levels of *Mvh*, *Oct4*, *Sod2*, *Gpx*, and *Cat* were detected after resveratrol treatment [94], implying that genes related to ovarian stem cell proliferation or anti-oxidative processes may help protect the ovary against chemotherapy-induced damage. An association between microRNA polymorphisms and the risk of premature ovarian insufficiency was also reported previously. Further investigations are warranted to identify significant protective genes against chemotherapy-induced ovarian damage.

### 4.2. Genetic Screening of Candidate Markers

Traditional biochemical markers for ovarian reserve include AMH level, follicle-stimulating hormone concentrations, inhibin-B level, and antral follicle count on ultrasound [7]. However, due to the development of genetic testing, several candidate genes for ovarian insufficiency are being investigated [85]. *Fmr1* and *Brca* testing can be performed easily in genetic clinics. Patients with mutations in these genes are at a higher genetic risk at baseline [95]. Evaluation of other frequent genetic variants, including *Nobox*, *Figla*, *Bnc1*, *Sohlh1*, *Sohlh2*, *Foxo3*, and *Hfm1*, could help identify individuals with increased genetic risk of ovarian damage due to chemotherapy. Next-generation sequencing could be considered in ovarian reserve testing by using targeted gene panels, whole-exome sequencing, or whole-genome sequencing [96]. The application of this technique is the future of genetic evaluation of patients who are at high risk of ovarian dysfunction after chemotherapy.

### 4.3. Other Options for Prevention of Ovarian Damage

#### 4.3.1. Gonadotropin-Releasing Hormone (GnRH)

Ovarian suppression using GnRH agonists before or during chemotherapy has protective effects on the ovaries by regulating the secretion of FSH and luteinizing hormone [97]. Ovarian suppression with this method protects ovarian function in young patients treated for lymphoma, breast cancer, and other diseases [98,99,100]. GnRH analogs have two possible theoretical mechanisms [101,102,103]. First, it involves decreasing the sensitivity of PFs entering the growing pool to gonadotoxicity. Furthermore, it constitutes the direct anti-apoptotic effect of GnRH agonists on ovarian germline stem cells. In combination with other modalities, the use of GnRH agonists, including oocyte or embryo freezing, may be a good option [104].

#### 4.3.2. AMH

In a previous study, the initiation of PF growth was inhibited when human ovarian cortical tissue was cultured with recombinant AMH [105]. Combining recombinant AMH with the cyclophosphamide metabolite in an ex vivo culture system maintained a high number of PFs in the ovaries [65]. AMH usually has limited activity in the ovaries, because it is an endogenous hormone.

#### 4.3.3. AS101

AS101 is a non-toxic immune modulator that acts on the PI3K/Akt/mTOR pathway [106]. AS101 was shown to diminish apoptosis in granulosa cells in an in vivo study [107]. AS101 was also related to a reduced follicle activation, thereby increasing follicle reserve and rescuing fertility after cyclophosphamide treatment [107].

#### 4.3.4. Imatinib

Imatinib is a tyrosine kinase inhibitor that selectively inhibits the ABL kinase domain of the bcr-abl oncogenic protein [108]. As PF depletion is mainly mediated by the ABL/TAp63 and PI3K/Akt/mTOR pathways, imatinib might prevent ovarian dysfunction caused by these pathways [109]. Many studies have investigated the protective effects of imatinib, but conflicting results have been reported [110,111,112].

#### 4.3.5. Sphingosine-1-Phosphate

Sphingosine-1-phosphate (S1P) inhibits the ceramide-promoted apoptotic pathway by increasing vascularity and angiogenesis and reducing PF apoptosis [72,113]. Co-administration of S1P with cyclophosphamide and doxorubicin was associated with a lower rate of apoptosis in mice [114]. It also showed a protective effect in mice treated with dacarbazine [115]. However, contradictory results were also reported in another study [116].

### 4.4. Cryopreservation

#### 4.4.1. Embryo Cryopreservation

Embryo cryopreservation is the most well-established method for preserving fertility [117]. Embryo freezing should be considered in patients who desire fertility preservation if there is adequate time for ovarian stimulation and if a partner or donor sperm is available [118]. Previous studies have demonstrated that embryo vitrification methods are better than slow freezing in pregnancy and live birth rates [119,120,121]. This option is not adequate for prepubertal girls because it requires ovarian stimulation. In studies comparing the fertilization and live birth rates of in vitro fertilization and embryo cryopreservation in patients with cancer, contradictory results were observed [122,123,124,125].

#### 4.4.2. Oocyte Cryopreservation

Oocyte cryopreservation is also considered a standard technique for fertility preservation in adeolescents and young adults with cancer [126]. The development of freezing techniques in assisted reproductive techniques has improved oocyte cryopreservation outcomes similar to those obtained with fresh oocytes [127,128]. It can also be utilized for women who are unmarried or do not want sperm donation. Vitrification was more effective than slow freezing in reducing cellular damage and chilling injury during the freezing process [129,130]. The combination of oocyte cryopreservation and ovarian tissue cryopreservation can enhance fertility [131].

#### 4.4.3. Ovarian Tissue Cryopreservation and Transplantation

Ovarian tissue cryopreservation could be considered for fertility preservation in children or young patients with cancer who need immediate treatment and do not have enough time for ovarian stimulation. Using this technique, a large number of oocytes can be preserved, and the hormonal functions of the ovary can be protected [132]. Slow freezing has been established as the preferred method for ovarian tissue cryopreservation rather than vitrification [133]. Ovarian activity was restored in 92.9% of the cases after transplantation of cryopreserved ovarian tissue by using the slow-freezing method [134]. Owing to the possible contamination of the ovarian tissue with malignant cells, this procedure is not utilized for patients with ovarian or hematologic malignancies [135,136].

## 5. Future Perspectives

The mechanism of chemotherapy-induced ovarian damage is not completely understood. Several studies have demonstrated that genes related to apoptosis or accelerated follicle activation are related to ovarian insufficiency. However, contradictory results have been reported. Validation of genetic profile screening for estimating susceptibility to chemotherapy-related ovarian damage may be further warranted. Patients with genetic variants involved in DNA repair or follicle activation can be screened before the initiation of cancer treatment via genetic testing. The establishment of genetic screening for fertility preservation could be helpful for young patients with cancer. Various other options are still under investigation.

Whole-ovarian transplantation has the benefit of immediate revascularization following blood vessel anastomosis [137,138]. Successful whole-ovarian cryopreservation and transplantation have been reported in animal studies [139,140,141,142]. However, potential injury due to hypothermic damage to blood vessels and difficulties in dispersing enough cryoprotective agents make it challenging in clinical practice.

In vitro maturation (IVM) can be used in patients with cancer who lack adequate time for ovarian stimulation or prepubertal girls who need immediate treatment. This requires immature oocyte retrieval and cryopreservation at an immature stage or a post-IVM mature state [143]. Several attempts have been reported; however, only a few live births have been reported after IVM procedures in patients with cancer [144,145,146].

Artificial ovaries can be useful for developing mature oocytes via in vitro culture of oocytes, isolated follicles, and ovarian tissue [147,148]. In animal models, this approach restored endocrine function, enabling in vivo follicle development and successful pregnancy; however, there have been no successful reports in humans [148,149]. Ovarian stem cells are under investigation for use in fertility preservation. Previous studies have reported successful detection and isolation of ovarian stem cells in animals and humans [150,151]. However, it is not commonly applied in clinical practice because of insufficient evidence in humans and ethical issues related to the use of oocytes and embryos [152]. Further studies are required to implement these approaches in clinical practice.

## 6. Conclusions

Fertility preservation in cancer patients is becoming more important; however, the effects on ovarian damage differ according to the type of agent and from patient to patient. Molecular mechanisms involved in cancer therapy-induced ovarian damage have been studied by many researchers. Understanding the molecular etiology of treatment-induced ovarian dysfunction can aid in identifying targets to prevent and reduce gonadal damage during cancer treatment and increase the number of options for fertility preservation.

The adoption of state-of-the-art genetic testing, including next-generation sequencing, has led to surprising developments in understanding the genomic aspects of ovarian insufficiency. The concept of precision medicine could be utilized to treat cancer and screen patients who have gonads vulnerable to chemotherapeutic agents, making it possible to plan individual fertility preservation options. As the genomic alterations of chemotherapy-induced ovarian damage continue to be investigated, mutations altering related molecular pathways may provide reliable information about reproductive potential.

Additionally, several novel therapies could be utilized in combination with standard FP techniques, or they may be used alone in the future. These strategies can assist young women who are not eligible for conventional methods because of their age or limited time before the initiation of disease treatment.

## Figures and Tables

**Figure 1 genes-12-01525-f001:**
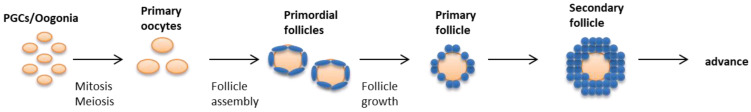
The morphogenesis of ovarian follicles.

**Table 1 genes-12-01525-t001:** Follicular development process and related genes.

Morphogenesis Process	Genes
From ovarian stem cell to ovigerous cords	*Wnt4, Rspo1, Bmp2, Bmp4, Bmp8a, Smads, KltL, Oct4, Nanos1, Nanos3, Kit, Dazl, Bcl-x, Pin1, Pog, Gja1, Bax*
Follicular assembly	*Msx1, Msx2, Dicer, Notch, Follistain, Stra8, Stag3, Syce1, Sycp2l, Spo11, Msh4, Msh5, Dmc1, Rec8, Atm, Lsh, Cdk2, Cbep, Hsf1, Fanc-a, Fanc-c, Fanc-g, Bcl2, Brca1, Mcl1, Gdf9, Bmp15*
Primordial follicles formation and growth	*Foxl2, Ccn2, Cyp19a1, KitL, p27kip1, Figla, Pdk1, Mcm8, Foxo3, Pten, Sohlh1, Sohlh2, Lhx8, Nobox8, Ybx2, Tsc1/2, Diaph2/3, Ngf, Nt3, Nt4, Bdnf, Ntrk1, Ntrk2, Ntrk3, Birc1, Becn1, Atg7*
Secondary follicle formation	*Gdf9, Bmp15, Zp1, Zp2, Zp3, Cx43, Rspo2, Inha, Taf4b*
Advance to latter stages	*Esr1, Fshr, Nppc, Nprb, Kras, Erk1/2, Egfr, Bax, Ahr, Clast4, Polg*

**Table 2 genes-12-01525-t002:** Ovarian damage with chemotherapeutic agents and their mechanisms of action.

Type of Chemotherapy	Agents	Target Disease	Mechanisms of Action
Alkylating agents	Cyclophosphamide Ifosfamide Nitrosoureas Chlorambucil Melphalan Busulphan Mechlorethamine	Leukemia, breast cancer, lung cancer, ovarian cancer, prostate cancer, lymphoma, Hodgkin’s disease	Interference with cell division via cross-linking of DNA; Mitochondrial transmembrane potential reduction; Inhibition of the accumulation of cytochrome *c* in the cytosol; Induction of DSBs in oocytes
Vinka alkaloids	Vinblastine Vincristine	Testicular cancer, lymphoma, Hodgkin’s disease, breast cancer, germ cell tumors, lung cancer,	Inhibition of tubulin forming into microtubules; Low gonadotoxic risk
Antimetabolites	Cytarabine Methotrexate 5-fluorouracil	Leukemia, breast cancer, ovarian cancer, gastrointestinal cancer	Inhibition of purine, pyrimidine becoming incorporated into DNA; Inhibition of RNA synthesis; Low gonadotoxic risk
Platinum agents	Cisplatin Carboplatin Oxaliplatin	Bladder cancer, colorectal cancer, head and neck cancer, lung cancer, ovarian cancer, testicular cancer	DNA damage by the formation of DNA adducts, which interfere with cellular transcription and replication, leading to oocyte death.
Anthracycline antibiotics	Daunorubicin Bleomycin Doxorubicin	Lymphoma, leukemia, breast cancer, sarcoma	Intercalation with DNA and prevention of its replication and transcription via the inhibition of topoisomerase II; Upregulation of P53 protein which induces apoptosis; DNA DSBs leading to activation of ATM, which initiates apoptosis
Others	Procarbazine	Hodgkin’s disease, brain tumor	Inhibition of DNA methylation and RNA and protein synthesis

DSB, double-strand breaks.

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
