# Peer review of "Genomic Consideration in Chemotherapy-Induced Ovarian Damage and Fertility Preservation"

_genes, 2021, doi:10.3390/genes12101525_

Round 1
Reviewer 1 Report
General
Methodology section is needed. Since this is a literature review, specific criteria as to how the literature was chosen (types of studies, years included, inclusion criteria, manuscripts excluded) needs to be included.
Introduction
The article discusses fertility preservation in both AYA and pediatric patients, given the discussion ovarian tissue cryopreservation/transplantation, and in vitro maturation
2.2 Germ Cell survival and DNA damage repair
Consider providing a more detailed mechanism of Fanc family genes in controlling ovarian reserve. Prior descriptions of genes within this paragraph provided a more exact mechanism. If specific mechanism is unclear, can move this gene down to final sentence of paragraph.
2.3 Follicular assembly and turnover
Line 95: “Deletion of this gene”
Clarify which gene this statement refers to or combine with prior sentence.
3.1 Chemotherapy-induced DNA DSBs
Line 113: Clarify the setting in which the p63 protein occurs. Is it a result of the failure of the repair pathway?
3.4 Genes related to chemotherapy-induced ovarian damage
Line 141-142: Consider detailing if the lower AMH levels, caused by Brca1 mutation, correlated to a reduction in pregnancy rate. Recommend elaborating further or stating that there needs to be more research performed.
4 Prevention Strategy for ovarian damage
Consider listing out types of cryopreservation (ie. Embryo, oocyte, ovarian tissue)
Consider detailing which of these options for fertility preservation are still considered experimental.
4.3.3 AS101
Line 217-218: Unclear if the “increased AMH levels” were increased from baseline or if the levels nadired during treatment then recovered to pre-treatment levels
4.4 Cryopreservation
While well organized, this section does not provide any type of genomic considerations within the context of cryopreservation. It appears disjointed from the purpose of the manuscript.
Author Response
I really appreciate for your review of the manuscript. I have applied all the changes you recommended. It was an honor for me to learn your professional point of view regarding detail approach for writing review article. I think this experience would improve my future works.
Thank you.
Best Regards,
Seongmin Kim
Methodology section is needed. Since this is a literature review, specific criteria as to how the literature was chosen (types of studies, years included, inclusion criteria, manuscripts excluded) needs to be included.
- Thank you for your kind advice. However, I do not think that methodology is necessary for this narrative literature review. The purpose of a literature review is to summarizes a topic that is broad in scope. There for it is not necessary defining what types of studies were included for review. Usually, systematic review defines a specific search strategy, and shows the inclusion and exclusion criteria of selected articles.
2.2 Germ Cell survival and DNA damage repair
Consider providing a more detailed mechanism of Fanc family genes in controlling ovarian reserve. Prior descriptions of genes within this paragraph provided a more exact mechanism. If specific mechanism is unclear, can move this gene down to final sentence of paragraph.
- Relevant mechanism of Fanc family genes are provided with reference.
2.3 Follicular assembly and turnover
Line 95: “Deletion of this gene”
Clarify which gene this statement refers to or combine with prior sentence.
- I have merged two sentences to clarify the meaning.
3.1 Chemotherapy-induced DNA DSBs
Line 113: Clarify the setting in which the p63 protein occurs. Is it a result of the failure of the repair pathway?
- Thank you for your advice. The role of P63 protein in female fertility was included in revised manuscript.
3.4 Genes related to chemotherapy-induced ovarian damage
Line 141-142: Consider detailing if the lower AMH levels, caused by Brca1 mutation, correlated to a reduction in pregnancy rate. Recommend elaborating further or stating that there needs to be more research performed.
- Thank you. I could not find any information about pregnancy rate regarding lower AMH levels caused by Brca However, IVF results were not different compared to non-Brca-carriers. And recommendation for further research was included.
4 Prevention Strategy for ovarian damage
Consider listing out types of cryopreservation (ie. Embryo, oocyte, ovarian tissue)
- Types of cryopreservation were already listed as 4.4.1-4.4.3 in original manuscript.
Consider detailing which of these options for fertility preservation are still considered experimental.
- Several established options including embryo cryopreservation and oocyte cryopreservation are already in clinical use. However, there are also experimental options including ovarian tissue cryopreservation, oocyte in vitro maturation, artificial ovary, and stem cell technologies.
4.3.3 AS101
Line 217-218: Unclear if the “increased AMH levels” were increased from baseline or if the levels nadired during treatment then recovered to pre-treatment levels
- The sentence has been revised to make the meaning clear.
4.4 Cryopreservation
While well organized, this section does not provide any type of genomic considerations within the context of cryopreservation. It appears disjointed from the purpose of the manuscript.
- Thank you for your opinion. However, the authors think that this section is needed to show the readers about the possible fertility preservation option as many as possible.
Reviewer 2 Report
In the current review, the authors have gathered information about the effects of chemotherapy on damages to the ovaries and fertility. This is an interesting and translational review, however, there are a few points that need to be addressed by the authors:
1-Please avoid unnecessary abbreviations such as AYAs, etc.
2- Table 1, please add relevant references to Table 1. The same with Table 2.
3-Lines 140-148, human genes should be capital and italicized here and throughout the manuscript.
4-Although this review is about the effects of chemotherapies on damages to the ovaries and the risk of infertility, it may be interesting and more comprehensive if the authors briefly include the possible effects of the PARP inhibitors as well.
Author Response
I really appreciate for your review of the manuscript. I have applied all the changes you recommended. It was an honor for me to learn your professional point of view regarding detail approach for writing review article. I think this experience would improve my future works.
Thank you.
Best Regards,
Seongmin Kim
1-Please avoid unnecessary abbreviations such as AYAs, etc.
- I have deleted unnecessary abbreviations.
2- Table 1, please add relevant references to Table 1. The same with Table 2.
- I have added relevant references in manuscripts for Table 1 and Table 2.
3-Lines 140-148, human genes should be capital and italicized here and throughout the manuscript.
- I have rechecked for all genes used in the manuscript, and revised following the review.
4-Although this review is about the effects of chemotherapies on damages to the ovaries and the risk of infertility, it may be interesting and more comprehensive if the authors briefly include the possible effects of the PARP inhibitors as well.
- Additional information about the negative effect of PARP inhibitors to ovarian function are briefly added in revised manuscript.